# Inhibitory Effects of Tangeretin, a Citrus Peel-Derived Flavonoid, on Breast Cancer Stem Cell Formation through Suppression of Stat3 Signaling

**DOI:** 10.3390/molecules25112599

**Published:** 2020-06-03

**Authors:** Yu-Chan Ko, Hack Sun Choi, Ren Liu, Ji-Hyang Kim, Su-Lim Kim, Bong-Sik Yun, Dong-Sun Lee

**Affiliations:** 1Interdisciplinary Graduate Program in Advanced Convergence Technology and Science, Jeju National University, Jeju 63243, Korea; koyuchan94@gmail.com (Y.-C.K.); liuren0308@gmail.com (R.L.); seogwi12@naver.com (J.-H.K.); ksl1101@naver.com (S.-L.K.); 2Subtropical/Tropical Organism Gene Bank, Jeju National University, Jeju 63243, Korea; choix074@jejunu.ac.kr; 3Division of Biotechnology, College of Environmental and Bioresource Sciences, Jeonbuk National University, Gobong-ro 79, Iksan 54596, Korea; bsyun@jbnu.ac.kr; 4Practical Translational Research Center, Jeju National University, Jeju 63243, Korea; 5Faculty of Biotechnology, College of Applied Life Sciences, Jeju National University, SARI, Jeju 63243, Korea

**Keywords:** tangeretin, cancer stem cells, Stat3, citrus, CD44^+^/CD24^−^, phytochemicals

## Abstract

Breast cancer stem cells (BCSCs) are responsible for tumor chemoresistance and recurrence. Targeting CSCs using natural compounds is a novel approach for cancer therapy. A CSC-inhibiting compound was purified from citrus extracts using silica gel, gel filtration and high-pressure liquid chromatography. The purified compound was identified as tangeretin by using nuclear magnetic resonance (NMR). Tangeretin inhibited cell proliferation, CSC formation and tumor growth, and modestly induced apoptosis in CSCs. The frequency of a subpopulation with a CSC phenotype (CD44^+^/CD24^−^) was reduced by tangeretin. Tangeretin reduced the total level and phosphorylated nuclear level of signal transducer and activator of transcription 3 (Stat3). Our results in this study show that tangeretin inhibits the Stat3 signaling pathway and induces CSC death, indicating that tangeretin may be a potential natural compound that targets breast cancer cells and CSCs.

## 1. Introduction

Breast cancer is the most frequently diagnosed cancer and the leading cause of cancer-related death among females [1]. Triple-negative breast cancer (TNBC) accounts for approximately 10–15% of all diagnosed breast cancers [2], and is defined as ER-, PR- and HER2-negative breast cancer. Women with TNBC have a high risk of recurrence within three years of diagnosis, and the mortality rate is increased for five years after diagnosis [3]. Cancer stem cells (CSCs) are relatively resistant to chemotherapy, radiotherapy and hormone therapy [4]. CSCs have functional roles in self-renewal and differentiation [5]. CSCs are responsible for the processes of cancer initiation, metastasis and cancer relapse [6]. Therefore, breast CSCs may contribute to drug resistance and relapse [7]. The CD44^+^/CD24^−^ being the most common cell-surface phenotype of breast CSCs can facilitate invasion, migration and proliferation [8].

Signal transducer and activator of transcription 3 (Stat3) plays a role in the inflammatory response and is a member of the seven-member Stat protein family (Stat1, 2, 3, 4, 5a, 5b, and 6) that is activated by growth factors [9]. Stat3 undergoes alternative splicing into Stat3α (92 kDa) and the isoform Stat3β (83 kDa). Stat3 plays a role in human oncogenesis. The activation of Stat3 signaling is associated with proliferation, antiapoptotic effects and cellular transformation [10,11]. Additionally, Stat3 plays a crucial role in differentiation under normal physiological conditions [6]. Inflammatory cytokines play key roles in regulating the interaction between CSCs and the IL6 inflammatory feedback loop, leading to the expansion of the CSCs population [12]. IL-8 interacts with the CXCR1 receptor on BCSCs (breast cancer stem cells), which promotes their cellular activities such as self-renewal and invasion. IL-8 signaling is a key pathway for regulating BCSCs [13].

Citrus species are natural products containing phytochemicals, which are promising for development into cancer therapies [14,15]. Citrus flavonoids, including nobiletin, hesperidin, tangeretin and naringin, have many biological activities, including a strong antioxidant and radical scavenging activity [16]. Tangeretin, also known as 4,5,6,7,8-pentamethoxyflavone, is a major compound in citrus peels. It has been shown to possess a variety of pharmacological activities, including anti-oxidative, anti-inflammatory and anticancer properties [17]. Inflammatory cytokine IL-6 plays an important role in mediating the interaction between CSCs and the microenvironment, which can influence tumor growth by regulating CSC subpopulations. However, no studies have shown the mechanisms underlying the targeted effects of tangeretin on CSC formation and Stat3 signaling in BCSCs. In our study, we showed that tangeretin had antiproliferative effects on breast cancer cells and reduced BCSC proliferation or prevalence through a decrease in Sox2 expression by inhibiting Stat3 signaling.

## 2. Results

### 2.1. CSC Inhibitor Derived from Citrus

A CSC-inhibiting compound derived from citrus was purified by bioassay-guided isolation, as shown in Figure 1A and Appendix A. The compound was isolated using organic solvent extraction, silica gel, gel filtration, TLC and preparatory HPLC. The isolated sample was determined to be a single compound using HPLC (Figure 1B). We assayed mammosphere formation using a purified sample (Figure 1C). The structural name of the purified compound was identified as tangeretin (Figure 2).

### 2.2. Tangeretin Suppresses the Proliferation of MDA-MB-231 and MCF-7 Cells and the Formation of Mammospheres

Breast cancer cells were incubated with various concentrations of tangeretin for 24 h. The antiproliferative function of tangeretin was assayed. Tangeretin inhibited cell proliferation, as shown in Figure 3A. To test whether tangeretin can suppress mammosphere formation, we treated cancer cells with tangeretin. Tangeretin inhibited mammosphere formation, as shown in Figure 3B. Tangeretin suppressed the migration and colony formation of cancer cell lines (Figure 3C,D). Our data indicated that tangeretin inhibited cancer hallmarks (proliferation, migration and colony formation) and mammosphere formation.

### 2.3. Tangeretin Modestly Induces Apoptosis in Mammospheres and Inhibits Mammosphere Proliferation

Late apoptosis in mammospheres was modestly induced by 80 μM of tangeretin (Figure 4A). Tangeretin reduced the transcript level of stem cell marker genes (Oct3/4, Sox2, and Nanog gene) (Figure 4B). To test whether tangeretin suppresses mammosphere growth, we treated mammospheres with tangeretin and counted the number of cancer cells derived from mammospheres. Tangeretin treatment inhibited mammosphere growth (Figure 4C). Our data showed that tangeretin, which disregulates the Stat3/Sox2 signaling pathway, was essential for inhibiting the proliferation of BCSCs.

### 2.4. Tangeretin Decreases Tumor Growth In Vivo

As tangeretin has antiproliferative effects, we examined whether tangeretin suppresses tumor formation in a nude mouse model. There was no significant body weight difference between control and tangeretin-treated mice (Figure 5A). At each time point, the tumor volume (Figure 5B) and weight (Figure 5C) of the tangeretin-treated nude mice were smaller than those of the untreated nude mice.

### 2.5. Tangeretin Treatment Modestly Reduces the CD44^+^/CD24^−^ Population Size

MDA-MB-231 and MCF-7 cells were treated with tangeretin, and we analyzed the BCSC marker-expressing CD44^+^/CD24^−^ subpopulation. Tangeretin modestly reduced the CD44^+^/CD24^−^ cell population size from 84.1% to 56.8% in MDA-MB-231 cells and from 2.2% to 0.9% in MCF-7 cells (Figure 6). This result shows that tangeretin reduces the frequency of a BCSC trait.

### 2.6. Tangeretin Inhibits the Stat3 Signaling Pathway and Reduces the Sox2 Level in Mammospheres

To investigate the cellular mechanism by which tangeretin inhibits mammosphere formation, we assessed the expression levels of Stat3 and pStat3 in mammospheres. Our results showed that tangeretin decreased the total protein levels of Stat3 and pStat3 in BCSCs (Figure 7A). The protein level of phospho-Stat3 was significantly reduced in the cytosol and nucleus of mammosphere cells. The Stat3 protein level was also decreased, as shown in Figure 7B. Additionally, we investigated Stat3 probe DNA binding to tangeretin-treated nuclear extracts by EMSA. We examined Stat3 probe DNA binding to mammosphere nuclear proteins using a Stat probe. Tangeretin reduced Stat3 DNA binding (Figure 7C, # 3). The specificity of Stat3 binding was determined using a self-competitor (100×) (Figure 7C, # 4) or a mutated Stat oligo (100×) (Figure 7C, # 5). The band indicated by arrows is Stat3 and the specific DNA complex. To examine the effect on Stat3 on mammosphere formation, we performed mammosphere formation using siRNA of Stat3. Our data showed that Stat3 reduction decreased mammosphere formation (Figure 7D). To analyze the cellular function of tangeretin, after tangeretin treatment, we checked the Sox2 level because Stat3 dimer activated Sox2 gene [18]. Our data showed that tangeretin reduced the level of transcripts and protein of Sox2 (Figure 7E). Sox2 plays a role in CSC progression [19]. Our data showed that the Stat3/Sox2 signal is important for mammosphere formation. Our data showed that tangeretin, which disregulates the Stat3/Sox2 signaling pathway, was essential for inhibiting the proliferation of BCSCs (Figure 8).

## 3. Discussion

It has been postulated that high consumption of fruits can prevent more than 20% of all cancer cases [20]. This preventive effect is predominantly mediated by phytochemicals interacting with specific target proteins that play important roles in cancer [21,22,23]. Citrus, one of the most important food sources of phytochemicals with health benefits, has many biological properties and controls key pathways involved in pathologies such as cancer [24,25,26]. The combination CD44^+^/CD24^−^ has emerged as the most important marker for BCSC isolation, and the CD24 population of MDA-MB-231 is low [27].

First, we purified a CSC inhibitor from citrus. Assay-based fractionation and several chromatographic methods isolated one compound from a citrus powder, tangeretin. Tangeretin is the major flavonoid of citrus. It also has antioxidant, anti-inflammatory and anticancer properties [17]. Tangeretin modestly induces apoptosis in bladder cancer cells through mitochondrial dysfunction [28]. Tangeretin and nobiletin induces G1 cell cycle arrest but not apoptosis in breast and colon cancers [29]. Nobiletin inhibits CD36-dependent tumor angiogenesis, migration, invasion and sphere formation through the CD36/Stat3/Nf-Κb signaling axis [30,31]. Quercetin suppresses breast CSCs through its inhibition of the PI3K/Akt/mTOR signaling pathway [32]. Despite numerous studies, there are no studies on tangeretin-induced antiproliferative and anti-CSC effects. Our results showed that tangeretin suppresses the proliferation of BCSCs. Tangeretin inhibited mammosphere formation in breast cancer cells (Figure 3) and modestly induced apoptosis in CSCs (Figure 4).

Our data showed the reduction of CD44^+^/CD24^−^ subpopulation, mammosphere formation, colony formation and tumor formation. It is known that tangeretin did not inflict damaging effects sufficient to result in a reduced capacity to survive and proliferate. However, inhibition of the growth of breast cancers without inducing cancer cell death may be advantageous in treating breast tumors. Breast cancer cells resumed growth similar to untreated control within a day of tangeretin removal [29]. The Stat3 protein is essential for the maintenance of BCSCs [4]. The acetylated derivative of tangeretin (5-AcTMF) had anticancer effects on human glioblastoma multiforme cells through blockade of Stat3 signaling [33]. Extracellular IL-8 protein is a factor for BCSCs formation [13]. We investigated Stat3 signaling under tangeretin treatment. Tangeretin suppressed the total protein levels of Stat3 and pStat3. The nuclear protein levels of Stat3 and pStat3 were also decreased by tangeretin. We assessed Sox2 transcript levels in BCSCs under tangeretin treatment and confirmed that the Sox2 mRNA level was decreased in the treated samples. In addition, the protein level of Sox2 was decreased in treated cells compared with control cells (Figure 7). Finally, tangeretin had an inhibitory effect on tumor growth in a breast cancer xenograft model. The tangeretin-treated group had a smaller tumor size and lower tumor weight than the control group.

In our study, tangeretin inhibited BCSC formation and targeted BCSCs by inhibiting the Stat3/Sox2 signaling pathway. Tangeretin is a possible therapeutic agent for breast cancer and BCSCs.

## 4. Materials and Methods

### 4.1. Reagents

Silica gel 60 and TLC plates were purchased from MERK (Darmstadt, Germany) and Sephadex LH-20 was obtained from Pharmacia (Uppsala, Sweden). High-pressure liquid chromatography was performed with a Shimadzu application system (Shimadzu, Kyoto, Japan). Cell viability was measured using the EZ-Cytox Cell Viability Assay Kit (DoGenBio, Seoul, Korea). Tangeretin was obtained from ChemFaces Co. (Hubei, China).

### 4.2. Plant Materials

Citrus peel was collected from Jeju Island, South Korea. The citrus peel was freeze-dried, and the dried citrus was ground. The samples (No. 2017_030) were deposited in the Department of Biomaterials, Jeju National University, JeJu-Si, South Korea.

### 4.3. Extraction and Isolation of an Inhibitor

Citrus powder was extracted with 100% methanol. The bioassay-based isolation protocol is summarized in Figure 1A. The methanol extracts were mixed with water, and the methanol was evaporated. The water-suspended samples were extracted with equal volumes of ethyl acetate. The ethyl acetate-concentrated part was loaded onto a silica gel column (3 × 35 cm) and fractionated with solvent (chloroform-methanol, 20:1) (Appendix A). The three parts were divided and assayed by evaluating mammosphere formation. The #2 part potentially inhibited mammosphere formation. The #2 part was loaded onto a Sephadex LH-20 open column (2.5 × 30 cm) and eluted in three fractions (Appendix A). The three parts were obtained and assayed by evaluating mammosphere formation. Part # 3 showed inhibition of mammosphere formation. Part #3 was isolated using preparatory TLC (glass plate; 20 × 20 cm) and developed in a TLC glass chamber. Individual bands were separated from the silica gel plate. Each fraction was assayed by evaluating mammosphere formation (Appendix A). The #1 fraction was loaded onto a Shimadzu HPLC instrument (Shimadzu, Tokyo, Japan). HPLC used an ODS 10 × 250 mm C18 column (flow rate; 3 mL/min). The mobile phase was water and acetonitrile. For elution, the acetonitrile proportion was initially set at 20%, increased to 80% at 20 min and finally increased to 100% at 40 min (Appendix A).

### 4.4. Structural Analysis of the Purified Sample

The chemical structures of the isolated compounds were determined by mass and nuclear magnetic resonance (NMR) measurements. The molecular weight was established as 372 by ESI-mass spectrometry, which showed a quasimolecular ion peak at *m/z* 373.3 [M + H]^+^ in the positive mode (Appendix A). The ^1^H NMR spectrum in CDCl_3_ exhibited signals due to four aromatic methine protons at δ 7.86 (2H) and 7.01 (2H), which are attributable to a 1,4-disubstituted benzene, one aromatic singlet methine at δ 6.59, and five methoxy groups at δ 4.09, 4.01, 3.94, 3.93 and 3.87. In the ^13^C NMR spectrum, there were 20 carbon peaks, including a carbonyl carbon at δ 177.3; nine sp^2^ quaternary carbons at δ 162.3, 161.2, 151.4, 148.4, 147.7, 144.0, 138.0, 123.8 and 114.8; five sp^2^ methine carbons at δ 127.7 (×2), 114.5 (×2) and 106.7; and five methoxy carbons at δ 62.2, 62.0, 61.8, 61.6 and 55.5 (Appendix A). All proton-bearing carbons were assigned by the HMQC spectrum, and the ^1^H-^1^H COSY spectrum established a partial structure, 1,4-disubstituted benzene (Appendix A). The chemical structure was determined to be from the HMBC spectrum, which exhibited long-range correlations from the methine proton at δ 6.59 to the carbons at δ 177.3, 161.2, 123.8 and 114.8, and from the methine protons at δ 7.86 to the carbon at δ 161.2, implying that this compound was a polymethoxylated flavone (Appendix A). Finally, long-range correlations from the five methoxy proton peaks to the oxygenated sp^2^ quaternary carbons established the structure of this compound as that of tangeretin (Figure 2).

### 4.5. Culture of Breast Cancer Cells and Mammospheres

Two breast cancer cell lines, MCF-7 and MDA-MB-231, were purchased from the American Type Culture Collection (Rockville, MD, USA) and maintained in DMEM supplemented with 10% fetal bovine serum (FBS; HyClone Fisher Scientific, CA, USA) and 1% penicillin/streptomycin (Gibco, Thermo Fisher Scientific, CA, USA). Cancer cells (5 × 10^4^ or 1 × 10^4^) were incubated in an ultralow-attachment 6-well plate with MammoCult^TM^ culture medium (STEMCELL Technologies, Vancouver, BC, Canada). All cells were cultured in a humidified 5% CO_2_ incubator at 37 °C for 7 days. The formation of mammospheres was assessed by the NICE program [34]. Mammosphere formation was examined by examining mammosphere formation efficiency (MFE) (%) [35].

### 4.6. Cell Viability Assay

MDA-MB-231 cells (2 × 10^4^ cells/well) were seeded in a 96-well plate for 24 h and treated with various concentrations (0, 5, 10, 20, 40, 60 and 80 μM) of tangeretin for 24 h in a medium with 10% FBS (proliferation assay). Then, we followed the manufacturer’s protocol for the EZ-Cytox Kit (DoGenBio, Seoul, Korea). The OD at a wavelength of 460 nm was measured using a GloMax^®^ Explorer microplate reader (Promega, Madison, WI, USA).

### 4.7. Colony Formation Assay

MDA-MB-231 and MCF-7 cells (2 × 10^3^ cells/well) were seeded in a 6-well plate, treated with different concentrations of tangeretin in DMEM and maintained for 7 days at 37 °C in a 5% CO_2_ incubator. The grown colonies were washed with 1× PBS three times, fixed for 10 min using 3.7% formaldehyde, treated for 20 min and stained for 20 min with 0.05% crystal violet. 

### 4.8. Wound-Healing Assay

MDA-MB-231 cells were seeded in a 6-well plate at 2 × 10^6^ cells/well. A scratch was made by using a microtip after the cells had grown into a monolayer. After the cells were washed two times with 1× PBS, the cancer cells were cultured with tangeretin in fresh DMEM for 24 h. Photomicrographs of the wounded areas were acquired using a light microscope [36].

### 4.9. Flow Cytometry Analysis 

After incubating with tangeretin for 24 h, cancer cells were dissociated using 1× trypsin/EDTA. We used a previously described method [36]. In total, 1 × 10^6^ cells were cultured with anti-CD44-FITC and anti-CD24-PE antibodies (BD, San Jose, CA, USA) on ice for 30 min. The cancer cells were centrifuged and washed two times with 1× FACS buffer and analyzed on an Accuri C6 (BD, San Jose, CA, USA).

### 4.10. Gene Expression Analysis 

Total RNA was extracted from cancer cells and purified, and real-time RT-qPCR was performed using a real-time One-step RT-qPCR kit (Enzynomics, Daejeon, Korea). We used a previously described method [37]. The specific primers are listed in Appendix A. 

### 4.11. Western Blot Analysis

Protein samples were extracted from mammospheres and cancer cells. After electrophoresis on a 12% SDS-PAGE gel, proteins were transferred to polyvinylidene fluoride (PVDF) membranes (Millipore, Burlington, MA, USA). The membranes were blocked in Odyssey blocking buffer at room temperature for 1 h and then incubated overnight with primary antibodies. The antibodies were anti-phospho-Stat3, anti-lamin B (Cell Signaling, Danvers, MA, USA), anti-Stat3, anti-Sox2 and anti-β-actin (Santa Cruz Biotechnology, Dallas, TA, USA). After membranes were washed three times using Tris-buffered saline/Tween 20, all membranes were incubated with IRDye 680RD- and IRDye 800W-labeled secondary antibodies for 1 h at room temperature, and the signal images were determined with an Odyssey CLx (LI-COR, Lincoln, NE, USA).

### 4.12. Electrophoretic Mobility Shift Assays (EMSAs)

Nuclear proteins were prepared as described previously [38]. EMSAs were performed with an Odyssey Infrared EMSA kit (LI-COR, Lincoln, NE, USA) according to the manufacturer’s instructions. IRDye 700-labeled forward and reverse strands of the Stat3 oligonucleotide were annealed. The IRDye 700-labeled Stat3 oligonucleotide was incubated with nuclear extracts in a final volume of 20 µL at room temperature for 20 min. The samples were electrophoresed on a 6% polyacrylamide nondenaturing gel, and EMSA data were visualized on an ODYSSEY CLx system (LI-COR, Lincoln, NE, USA).

### 4.13. Xenograft Transplantation

Twelve female nude mice were injected with two million MDA-MB-231 cells with or without an additional tangeretin (2.5 mg/kg) injection. Tumor volume was estimated for 35 days using the formula (width^2^ × length)/2. The mouse experiments were performed as described previously [39]. Animal care and animal experiments were performed in accordance with protocols approved by the Institutional Animal Care and Use Committee (JNU-IACUC; Approval Number 2017-003) of Jeju National University. Female nude mice (4 weeks old) were purchased from OrientBio (Daejeon, South Korea) and cultured in mouse facilities for 1 week.

### 4.14. SiRNA of Stat3

To examine the inhibitory function of Stat3 on mammosphere formation, breast cancer cells were transfected with siRNAs targeting human Stat3 (Bioneer, Daejeon, South Korea). The Stat3 siRNAs (NM_1145658) and a scrambled siRNA were purchased from Bioneer (Daejeon Cor., South Korea). For siRNA transfection, cancer cells were cultured and transfected using Lipofectamine 2000 (Invitrogen, Carlsbad, CA, USA) according to the manufacturer’s instructions. The protein levels of Stat3 were determined via immunoblot analysis.

### 4.15. Statistical Analysis

All data were analyzed with GraphPad Prism 7.0 software (GraphPad Prism, Inc., San Diego, CA, USA). All data from three independent experiments are reported as the mean ± standard deviation (SD). Data were analyzed using one-way ANOVA. A *p*-value of less than 0.05 was considered significant.

## 5. Conclusions

A CSC-inhibiting compound from citrus extracts was purified using silica gel, gel filtration, TLC, and HPLC. The compound was identified as tangeretin. Tangeretin inhibited cell proliferation, CSC formation and tumor growth, and modestly induced apoptosis in CSCs. The size of the CSC subpopulation (CD44^+^/CD24^−^) was reduced by tangeretin. Tangeretin reduced the total level and phosphorylated nuclear level of Stat3 Tangeretin decreased the transcript levels of Sox2 and Sox2 protein in mammospheres. Our results in this study show that tangeretin inhibits the Stat3/Sox2 signaling pathway and induces CSC death, indicating that tangeretin may be a potential natural compound targeting breast cancer.

## Figures and Tables

**Figure 1 molecules-25-02599-f001:**
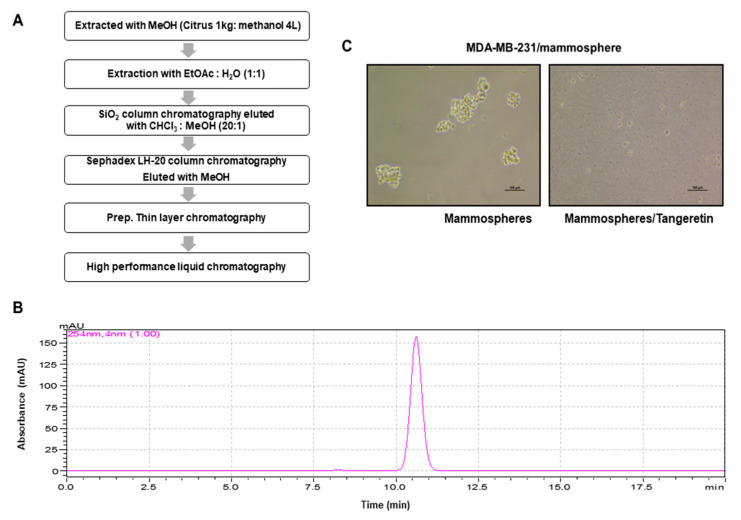
Purification protocol for a CSC inhibitor derived from citrus peels and a mammosphere formation assay using a purified sample. (**A**) Flowchart for the isolation of the mammosphere inhibitor. (**B**) HPLC chromatogram of the inhibitor purified from citrus. (**C**) Assay for mammosphere formation in the presence of the HPLC-purified sample. Cancer cells were treated with the HPLC-purified sample. Images show representative mammospheres, and were imaged by microscopy (scale bar: 100 μm).

**Figure 2 molecules-25-02599-f002:**
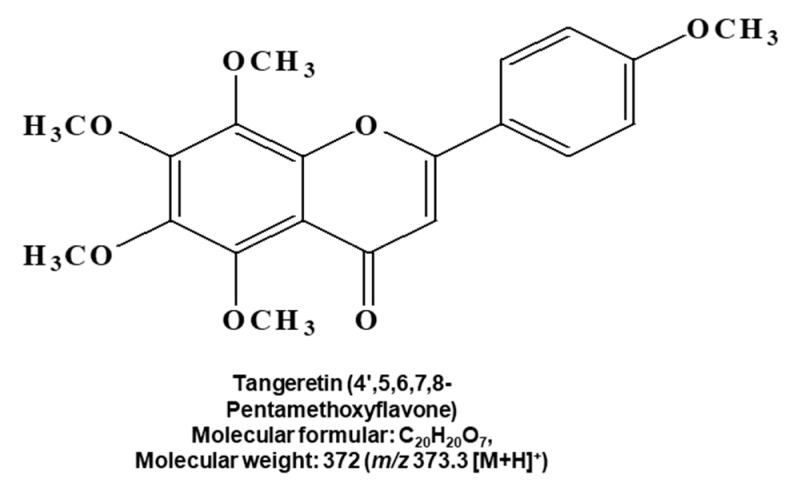
Chemical structure of the compound purified from citrus. Chemical structure of tangeretin.

**Figure 3 molecules-25-02599-f003:**
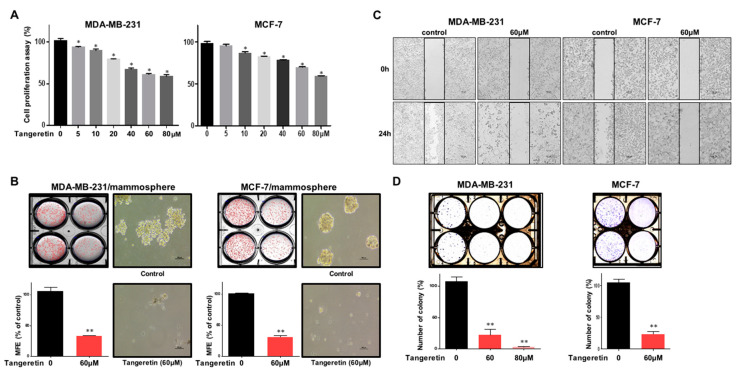
The effects of tangeretin on cell proliferation and mammosphere formation. (**A**) MDA-MB-231 and MCF-7 cells were treated with tangeretin for 24 h in a medium supplemented with 10% FBS. The cytotoxicity of tangeretin was measured with an EZ-Cytox kit. (**B**) Tangeretin inhibits the formation of mammospheres. To establish mammospheres, 1 × 10^4^ MDA-MB-231 cells and 5 × 10^4^ MCF-7 cells were seeded in ultralow-attachment 6-well plates with CSC culture medium. The mammospheres were incubated with increasing concentrations of tangeretin. Photos show representative mammospheres, and were captured by microscopy (scale bar: 100 μm). Mammosphere formation efficiency (MFE) was examined. (**C**) The effect of tangeretin on the migration of breast cancer cells (MDA-MB-231 and MCF-7 cells) was evaluated. Migration with or without tangeretin was captured at 0 and 24 h (scale bar: 100 μm). (**D**) Tangeretin inhibits colony formation by cancer cells. Breast cancer cells (MDA-MB-231 and MCF-7 cells) were incubated and treated with tangeretin. Representative data were collected. The data from triplicate experiments are represented as the mean ± SD; * *p* < 0.05, ** *p* < 0.01.

**Figure 4 molecules-25-02599-f004:**
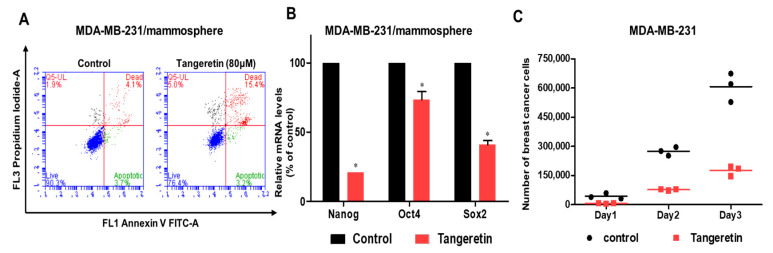
The effects of tangeretin on apoptosis and mammosphere growth. (**A**) Tangeretin modestly induced apoptosis in mammospheres treated with tangeretin. We induced mammosphere formation and treated mammospheres with tangeretin. After treatment, apoptotic cells were examined using Annexin V/PI staining. (**B**) Transcription levels of CSC markers, including the Nanog, Sox2 and Oct4 genes, determined in tangeretin- and DMSO-treated mammospheres using CSC marker-specific primers and real-time PCR. β-actin acts as an internal control. The data shown represents the mean ± SD of three independent experiments. * *p* < 0.05 vs. DMSO-treated control. (**C**) Tangeretin inhibited mammosphere growth. Mammospheres with or without tangeretin were dissociated into single cells, and the single cells were plated in 6 cm dishes in equal numbers. The cells were examined one, two and three days later.

**Figure 5 molecules-25-02599-f005:**
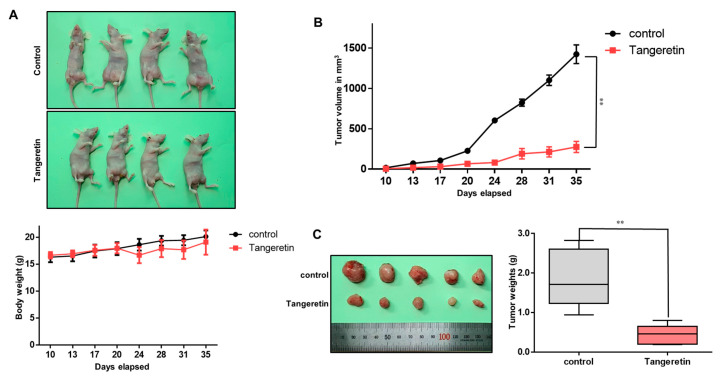
The effect of tangeretin on tumor growth in a xenograft model. (**A**) MDA-MB-231 cells (2 × 10^6^ cells/mouse) were inoculated into the mammary fat pad of female nude mice and treated with tangeretin (2.5 mg/kg) or DMSO (n = 6). Nude mice were intraperitoneally injected with/without Tangeretin (2.5 mg/kg) once a week for a total of four time injections. The body weights of the tangeretin-treated group were comparable to those of the control group. (**B**) Tumor volume was calculated as (width^2^ × length)/2 at the indicated time points. (**C**) The tumor weights of the control and tangeretin-treated mice were assayed after sacrifice at day 35. The data are presented as the mean ± SD of three independent experiments. ** *p* < 0.05 versus the DMSO-treated control group.

**Figure 6 molecules-25-02599-f006:**
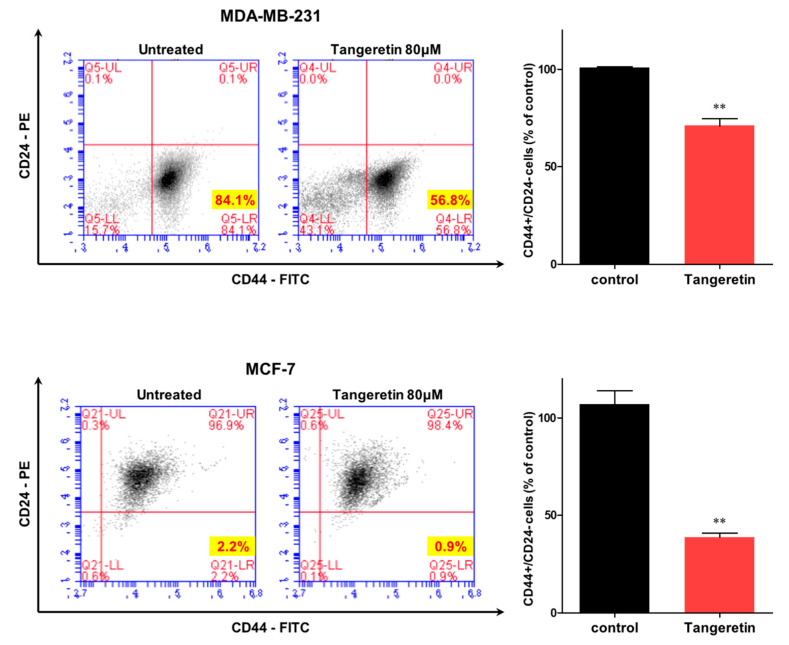
The effect of tangeretin on the CD44^high^/CD24^low^ cell proportion. The CD44^high^/CD24^low^ subpopulation within an MDA-MB-231 and MCF-7 cell population treated with tangeretin (80 μM) or DMSO for 24 h was analyzed by flow cytometry. For flow cytometry analysis, 2 × 10^4^ cells were acquired. The gating was based on the binding of an antibody without tangeretin (red cross). The data from triplicate experiments are represented as the mean ± SD; ** *p* < 0.01.

**Figure 7 molecules-25-02599-f007:**
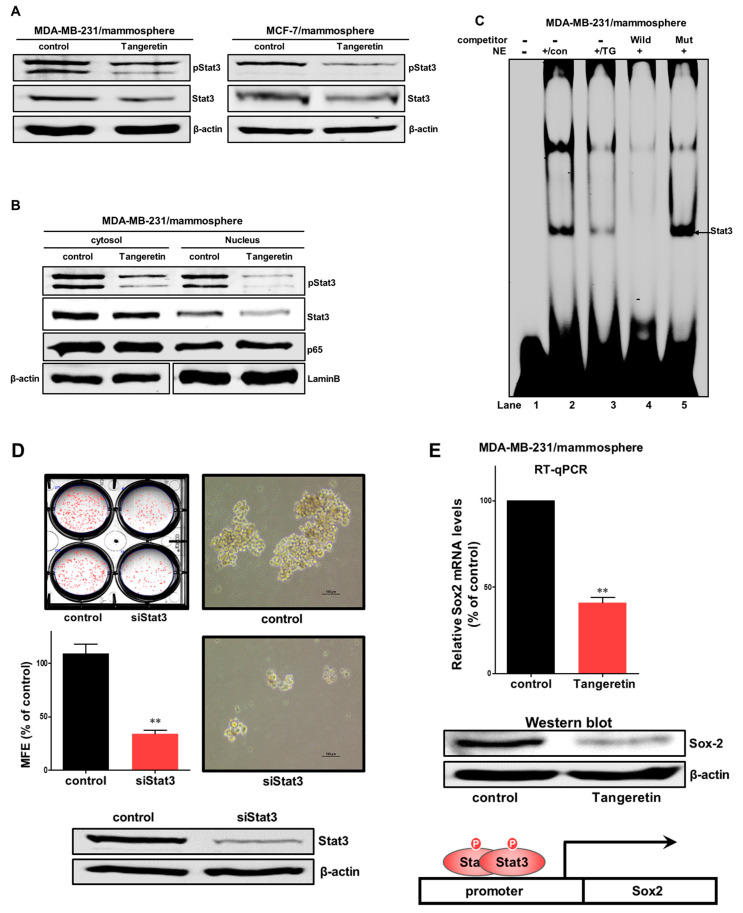
Tangeretin regulates Stat3 signaling and Sox2 regulation. (**A**) The total levels of Stat3 and pStat3 were measured in mammospheres composed of MDA-MB-231 cells and MCF-7 cells after treatment with tangeretin (0 or 60 μM) for 48 h using western blotting. Total lysates were subjected to immunoblot analysis with anti-Stat3 and anti-pStat3 antibodies. β-actin was used as an internal control. (**B**) The levels of Stat3 and pStat3 in the cytosolic and nuclear protein fractions were measured in mammospheres composed of MDA-MB-231 cells after treatment with tangeretin for 48 h using western blotting. Nuclear and cytosolic proteins were run on SDS-PAGE gels, followed by immunoblotting with anti-Stat3, anti-pStat3, anti-β-actin and anti-Lamin B antibodies. (**C**) EMSA was used to analyze mammosphere nuclear proteins after treatment with tangeretin. Nuclear extracts were reacted with a Stat3 probe and subjected to 6% native PAGE. Lane 1: Stat3 probe; lane 2: nuclear extracts with the Stat3 probe; lane 3: tangeretin-treated nuclear proteins with the Stat3 probe; lane 4: nuclear proteins incubated with a self-competitor oligo (100×); and lane 5: nuclear extracts incubated with a mutated-stat3 probe (100×). The arrow indicates the DNA/Stat3 complex in the mammosphere nuclear lysates. (**D**) Effect of Stat3 on mammosphere formation using siRNA of Stat3. Stat3-downregulated MDA-MB-231 cells were cultured for seven days using mammosphere media. Images were obtained by microscrope at 100× magnification. (**E**) The transcriptional level of the Sox2 gene in MDA-MB-231 was determined in tangeretin-treated mammospheres. A Sox2-specific primer was used for real-time RT-qPCR. Western blot analysis of mammosphere under tangeretin. β-actin served as an internal control. The data are presented as the mean ± SD of three independent experiments. ** *p* < 0.01 versus the DMSO-treated control group.

**Figure 8 molecules-25-02599-f008:**
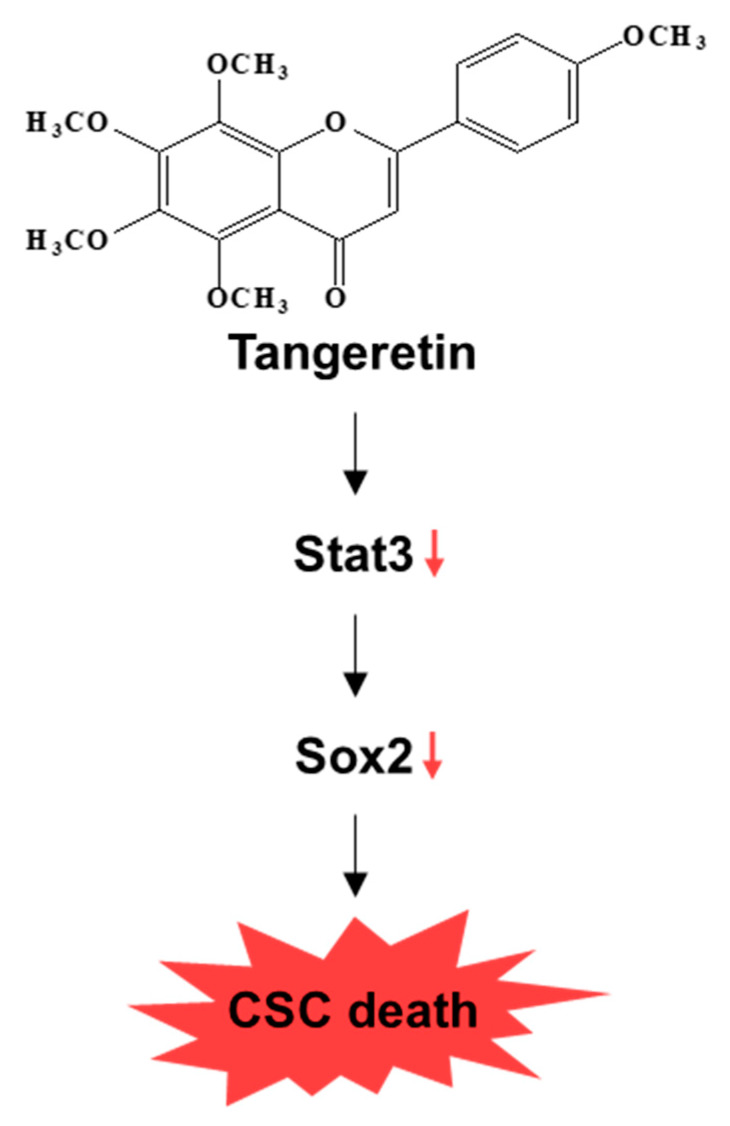
The proposed model for CSC death induced by tangeretin is shown.

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
