# Peer review of "Inhibitory Effects of Tangeretin, a Citrus Peel-Derived Flavonoid, on Breast Cancer Stem Cell Formation through Suppression of Stat3 Signaling"

_molecules, 2020, doi:10.3390/molecules25112599_

Round 1

Reviewer 1 Report

In this manuscript, Ko et al. described the inhibitory effect of Tangeretin on the stemness of breast cancer stem cells (BCSCs)-like cells derived from two human breast cancer cell lines MDA-MB-231 and MCF-7 at both in vitro and in vivo levels and demonstrated that this stemness-inhibitory effect of Tangeretin is achieved by suppression of STAT3 signaling. Overall, this study was excellently performed, results were clear-cut, and the conclusion was solidly supported by the evidence presented. A couple of minor comments are suggested herein to improve the quality of this article:

  1. For correcting the grammar mistake and highlighting the role of STAT3 signaling in the main conclusion of this story, the title of this article is recommended to be changed as “Inhibitory Effects of Tangeretin, A Citrus Peel-Derived Flavonoid, on Breast Cancer Stem Cell Formation through Suppression of STAT3 signaling”.
  2. The font and format of the “Stat protein” shown in the second line at page 2 must be corrected.
  3. Please define the abbreviated term “BCSCs” when it was first appeared in the manuscript.
  4. The decrease in CD44+/CD24- levels after Tangeretin treatment must be presented in a quantitative manner with statistical analysis.
  5. Results shown in Figures 7A-C are highly recommended to be organized as a new Figure 4, as these results describe Tangeretin-mediated inhibition of mammosphere formation and therefore are in line with the results shown in the original Figure 3. The original Figure 4 and ensuing Figures are therefore numbered consecutively started from Figure 5, and the original Figure 7D can be presented as an independent Figure.
  6. The inhibitory effect of Tangeretin on STAT3 signaling is quite noteworthy. In fact, STAT3 inhibition appears to be an important mechanism of action of Tangeretin and its structure-related compounds to exert their anti-cancer actions. For instance, an acetylated derivative of Tangeretin, 5-acetyloxy-6,7,8,4’-tetramethoxyflavone, was recently shown to be anti-glioblastoma mutiforme through blocking STAT3 signaling (Cheng YP et al. Blockade of STAT3 Signaling Contributes to Anticancer Effect of 5-Acetyloxy-6,7,8,4'-Tetra-Methoxyflavone, a Tangeretin Derivative, on Human Glioblastoma Multiforme Cells. Int J Mol Sci. 2019;20(13):3366). Thus, it is recommended to cite Cheng YP et al. as a supporting reference to underscore the notion of Tangeretin’s inhibitory effect on STAT3 signaling in the Discussion
  7. In the Materials and Methods section (page 11), all the bold fonts in subtitle 14 should be removed.

Author Response

We submit the Reviewer 1 comments.

Reviewer 2 Report

The paper by Ko et al. reports an investigation of the activity of tarengetin, a natural component of Citrus, against cancer stem cells, and concludes that tarengetin exerts its activity through the inhibition of STAT3 signaling pathway.

The proposed mechanism of cellular/biochemical effects of tangeretin is of potential imterest for therapeutic implications. However, in ths context, the critical question is related to the pharmacological relevance of the reported cellular observations.

Specifically:

a) the inhibitory effects of the tested compound have been observed in vitro at relatively high concentrations (60 microM). Are these concentrations achieved in vivo for therapeutic applications?

b) The reported experiments to support the in vivo efficacy ( Fig.4) do not provide adequate details concerning the treatment ( e.g., shedule, dose and route of drug administration)

These points should be clarified

Other comments:

The antiproliferative activity against MDA-MB-231 and MCF-7 cells (fig. 3) should be expressed in terms of IC50, in order to allow comparison with other active substances. Did the authors use e reference compound?

The amount of tarengetin the authors obtained by extraction, isolation and purification processes is not reported. The amount of compounds obtained in each single step should be detailed, together with the amount of starting material (dried citrus powder).  

Some preceding literature on the anticancer activity of tarengetin seems to have been overlooked (e.g Periyasamy, K et al.. . Cancer Chemother. Pharmacol. 2015, 75, 263–272, or the papers about the activity of 5-acetyl tarengetin)

Minor points:

Pag. 2: The authors state: Citrus species are natural products containing phytochemicals that are promising for development into cancer therapies The next sentence reports the same concept, thus it can be removed

Pag 2 Citrus flavonoids containing nobiletin, hesperidin, tangeretin, and naringin have many biological activities: Did the Authors  mean “including”?

Pag 4 The phrase: “We show that tangeretin regulates tumorigenicity in a xenograft mouse model” should be removed, the content is the same of the previous one

Pag 7“Tangeretin treatment inhibited mammosphere growth. Our data indicated that tangeretin reduced mammosphere growth”  Same as before

Pag 8“Tangeretin induces G1 cell cycle arrest but not apoptosis in breast and colon cancers [29]. Nobiletin inhibits sphere formation of breast cancer [30, 31]. Quercetin suppresses breast CSCs.” The concepts should be better explained

Pag 9 extracted with 1X ethyl acetate(?)

pag 9 The #2 fraction ?

Pag 11To examine the inhibitory function of Stat3 on mammosphere formation, breast cancer cells were transfected with ....remove bold style

Author Response

We submit the Reviewer 2 comments.
